# *Marrubium vulgare* L.: A Phytochemical and Pharmacological Overview

**DOI:** 10.3390/molecules25122898

**Published:** 2020-06-24

**Authors:** Milica Aćimović, Katarina Jeremić, Nebojša Salaj, Neda Gavarić, Biljana Kiprovski, Vladimir Sikora, Tijana Zeremski

**Affiliations:** 1Institute of Field and Vegetable Crops Novi Sad, Maksima Gorkog 30, 21000 Novi Sad, Serbia; biljana.kiprovski@ifvcns.ns.ac.rs (B.K.); vladimir.sikora@ifvcns.ns.ac.rs (V.S.); tijana.zeremski@ifvcns.ns.ac.rs (T.Z.); 2Department of Pharmacy, Faculty of Medicine, University of Novi Sad, Hajduk Veljkova 3, Novi Sad 21000, Serbia; katarina.jeremic@mf.uns.ac.rs (K.J.); nebojsa.salaj@mf.uns.ac.rs (N.S.); neda.gavaric@mf.uns.ac.rs (N.G.)

**Keywords:** bioactivity, horehound, medicinal properties, morphology, marrubin

## Abstract

*Marrubium vulgare* is a plant with high bioactive potential. It contains marrubiin, a labdane diterpene that is characteristic for this genus, as well as a complex mixture of phenolic compounds. According to numerous studies, *M. vulgare* acts as a good antioxidant agent, and due to this, it could potentially be useful in treatments of cancer, diabetes mellitus, and liver diseases. In addition, its anti-inflammatory, wound-healing, antihypertensive, hypolipidemic, and sedative potential are discussed. Apart from that, its antimicrobial activity, especially against Gram+ bacteria, fungi, herpes simplex virus, and parasites such as *Toxoplasma gondii*, *Trichomonas vaginalis*, and *Plasmodium berghei-berghei* was recorded. Additionally, it could be used as a chicken lice repellent, herbicide, and natural insecticide against mosquito larvae and natural molluscicide. In veterinary medicine, *M. vulgare* can be used as an anthelmintic against the eggs and larvae of bovine strongyles parasites, and as an antibiotic against bovine mastitis caused by resistant bacterial strains. Due to the mentioned benefits, there is a tendency for the cultivation of *M. vulgare* in order to ensure high-quality raw material, but more firm scientific evidence and well-designed clinical trials are necessary for the well-established use of *M. vulgare* herb and its preparations.

## 1. Introduction

*Marrubium vulgare* L. (*M. vulgare*), originated in the region between the Mediterranean Sea and Central Asia, has become a widespread species, currently inhabiting all continents [1]. The Latin name *Marrubium* derives from the Hebrew word *marrob*, meaning bitter juice, while *vulgare* means ‘common’ or ‘well known’. In the English language, the name ‘horehound’ comes from the Old English words *har* and *hune*, meaning downy plant. The most common folk name in the Serbian language is *očajnica* meaning a ’desperate woman’, because tea from this herb was typically used as a bitter remedy by women who were unable to conceive (for regulation of menstrual cycle). Nevertheless, the most prevailing traditional application of *M. vulgare* worldwide is for the treatment of gastrointestinal and respiratory disorders.

This review article is aimed at comprehensively describing the botanical features, phytochemicals, and pharmacological properties of *M. vulgare* along with highlighting the interest in this medicinal plant as a source of herbal remedies. This is supported by the traditional use of *M. vulgare* preparations in medicine as stated by the European Medicines Agency (EMA) commission [2]. It is based on the literature collected from SciFinder, ScienceDirect, Scopus, PubMed, Web of Science, and other sources. However, more than 80 publications available are included in this paper. For that reason, this review article could present a valuable update about current knowledge [3,4,5].

## 2. Botanical and Agronomical Features

*M. vulgare* is an annual or perennial herb from Lamiaceae family, with a tough, woody, branched taproot or numerous fibrous lateral roots and numerous stems, which are quadrangular, erect, very downy, and from 20 to 100 cm high. Leaves are roundish, ovate, usually toothed, petiolate, veined, and hoary on the surface, and they are arranged in opposite pairs on a long stem. Inflorescences are formed in axils of upper leaves, with white flowers in crowded axillary whorls. The calyx is tubular, lobed, and 10-toothed, each tooth with a small hooked spine/bristle. Corolla is white to pale lavender, tubular, and bilabiate; the upper lip is 2-lobed, bifid, and erect, while the lower lip is 3-lobed with a larger middle lobe. Corolla tube includes style, stamens, and anthers with divergent sacs [5]. The shape of pollen grains is oblate–spheroidal, radially symmetrical, and isopolar. Flowers in *M. vulgare* generally appear in the early spring, and they are regularly visited by nectar-gathering bees [4]. Seeds lie at the bottom of the calyx [3].

The surface of *M. vulgare* vegetative and generative organs is densely covered with glandular and non-glandular trichomes. There are two types of glandular trichomes: peltate and capitate. Peltate trichomes consist of a short stalk cell and a large head with secretory cells arranged in a circle. The substance produced by secretory cells passes through apical walls and accumulates within a space between the cuticle and the cell wall layer. The plant has a capitate, long trichomes, with basal cells, a long stalk neck cell, and a unicellular head. There are two types of short capitate trichomes: those with a bicellular head and those with a unicellular stalk. As for non-glandular trichomes, they can be multicellular uniserate or multicellular branched [6].

Bearing in mind its vast potential for medicinal use, as well as the continuous discovery of its alternative beneficial properties, lately there has been a growing interest and demand for *M. vulgare*. In order to secure the supply of good quality raw material with a high content of marrubiin and other diterpenes, as well as phenolics, the cultivation of this plant is becoming increasingly common. In addition to choosing the assortment and agricultural practices appropriate for specific agro-ecological conditions, the postharvest processing of raw material plays a key role.

*M. vulgare* is reproduced exclusively by seeds, through direct sowing or seedlings production. However, after the collection of mature seeds, its germination rate is low (35%). Following one month storage, germination increases to 78–80% [4]. Seeds sown in autumn germinate the following spring, whereas those sown in spring germinate after three weeks [7].

The cropping of *M. vulgare* grown from direct sowing in the field and seedlings production was assessed in the experiments conducted in Poland. It was established that the seed sowing term significantly affected *M. vulgare* cropping. Actually, a much higher fresh herb yield was obtained from the plantation established in the spring than from that established in autumn [7]. Harvest date significantly influenced *M. vulgare* cropping in the second year.

The biofertilization of *M. vulgare* plants with nitrogen biofertilizer (Biofert), potassium biofertilizer (Available K), or with a combination of the two showed that when applied abundantly, the fertilizers enable the *M. vulgare* plant to reach its maximum height, and also enhance fresh and dry herb weights and essential oil production [8]. Physiological analyses showed that an increase of the photosynthetic photon flux from 0 to 100 μmol/m^2^/s lead to significant linear, curvilinear, and quadratic effects on nitrogen, boron, and manganese, respectively, in aerial parts [9].

*M. vulgare* prefers alkaline soils [10]. Amri et al. [11] reported that different concentrations of copper reduce the uptake and translocation of cationic elements Fe^2+^, K^+^, and Ca^2+^ in a Cu^2+^ concentration-dependent manner in the aboveground part of the plant. Treatment with copper stimulates antioxidant enzymes activities (superoxide dismutase (SOD) and catalase (CAT) enzymes) and increases the total phenolics (flavonoid) content. These findings lead the authors to conclude that *M. vulgare* possesses an intrinsic capability to cope with the Cu stress by the activation of enzymatic and non-enzymatic antioxidant systems.

Other results [12] showed that salt treatment (different concentrations of NaCl) affects the morphological, physiological, and biochemical traits of *M. vulgare*, which appears to be highly sensitive to salinity, especially the content of the main bioactive compound marrubiin, which decreased with higher NaCl concentration.

Boron plays a very important role in sugar synthesis in cell walls, nucleic acid, phenolics, hormone, carbohydrate and protein metabolism, cell elongation and pollen tube formation in plants. However, the deficiency and toxicity levels of boron are quite close when it comes to plants, which need only trace amounts of boron for vital activities. Ardic et al. [13] used a curcumin method for the identification of boron content in *M. vulgare* plant specimens (root, stem leaf, and flower). In addition, soil samples were analyzed for boron content via the atomic absorption spectrophotometer technique. Their report confirmed that compared to the boron concentration in soil, *M. vulgare* samples accumulated boron levels that were 4 times higher in flowers, more than 4 times higher in leaves, 3 times higher in stem, and about 3 times higher in root, which showed that the members of the natural populations of *M. vulgare* can tolerate high boron stress.

## 3. Phytochemical Composition

*Marrubii herba* (flowering aboveground parts) are harvested just before obtaining a full green color. *M. vulgare* has a musky odor that changes by drying into a pungent yet pleasant odor and bitter, aromatic taste [3]. Mittal and Nanda [14] reported that *Marrubii herba* (dried to 17.20% of fresh herba weight) has a total fiber content of 9.50%, total ash content of 10.70%, acid insoluble ash content of 1.73%, and water-soluble ash content of 8.90%. They also reported that the alcohol soluble extractive value is 8.66%, which suggested that most of the constituents of the plants were soluble in alcohol. Furthermore, the water-soluble extractive value is approximately 5.90%, while the petroleum ether soluble extractive value is 2.77%.

*M. vulgare* produces trace amounts of essential oil, usually between 0.03% and 0.06% [7,8] with monoterpenes such as camphene, p-cymol, fenchene, limonene, α-pinene, sabinene, and α-terpinolene [2]. Non-volatile monoterpene derivatives are also present in the plant with monoterpene marrubic acid and monoterpene glycoside sacranoside A (myrtenyl 6-*O*-α-l-arabinopyranosyl-β-d-glucopyranoside) as identified compounds [15,16]. Sesquiterpene lactone vulgarin, β-sitosterole, lupeol, and β-amyrin types of triterpenoids such as oleanolic acid have been identified in *M. vulgare* extracts [17,18]. *M. vulgare* accumulates diterpenes of labdane type as principle bitter components, up to 3 mg/g of fresh weight [19], with marrubiin being the predominant one (0.12–1%) followed by its precursor pre-marrubiin (0.13%), 12(*S*)-hydroxymarrubiin, 11-oxomarrubiin, 3-deoxo-15(*S*)-methoxyvelutine, marrubenol, marruliba-acetal, cyllenil A, polyodonine, and preleosibirin [2,15,20,21]. In addition, peregrinol, peregrinin, dihydroperegrinin, vulgarol, vulgarcoside A, deacetylvitexilactone, carnosol, deacetylforskolin are present in diterpenoid fraction [2,15,17,20,21,22,23,24]. The estimated content of marrubiin in the methanol extract of *M. vulgare* is 156 mg/g [25]. The review of the main compounds usually quantified in *Marrubii herba* is presented in Table 1, while the list of individual non-volatile terpenoid compounds is presented in Table 2.

Marrubiin attributes include a low turnover, high stability, and little catabolism, which are core characteristics required for therapeutic compounds and nutraceuticals of economic importance. It is responsible for the therapeutic properties such as antinociceptive, antioxidant, antigentoxic, cardioprotective, vasorelaxant, gastroprotective, antispasmodic, immunomodulating, antioedematogenic, analgesic, and antidiabetic properties. In addition, marrubiin is considered a potential substrate for potent active compounds, particularly marrubiinic acid and marrubenol. Furthermore, marrubiinic acid possesses antinociceptive activity, while marrubenol is vasorelaxant [35].

Furthermore, *M. vulgare* is an abundant source of various phenolic compounds, major classes being phenolic acids, phenylpropanoid (cinnamic) acids, and esters and flavonoids. The amount of total cinnamic acid derivatives is estimated to 14.09 mg/100 mg of dry material [27]. *M. vulgare* contain tannins (up to 7%) where the estimated amount of condensed tannins is 16.55 mg catechin/100 g of dry material [2,26]. Detailed chemical analysis identified gallic, gentisic, p-hydroxybenyoic, protocatechuic, and syringic acids. In the cinnamic acids subgroup of phenolic compounds, *trans*-cinnamic, ferulic, o-coumarinic, p-coumarinic and sinapic acids are determined. Chologenic acid and other cinnamic acid esters were quantified. It was considered that as a part of the Lamioideae subfamily, *M. vulgare* is characterised by bitter components, a low amount of essential oil, and an absence of rosmarinic acid [2,38]. However, novel data identified rosmarinic acid in *M. vulgare* extracts [18,22,39]. Hydroxycinnamic acid derivatives, phenylethanoid glycosides, such are acteoside (used as qualitative marker), alyssonoside, arenanoside, ballotetroside, marruboside, and others represent another subgroup of phenolic compounds present in *M. vulgare* extracts. The list of individual bioactive phenolic acids and esters is given in Table 3.

In flavonoid fractions, flavone aglycones such as apigenin, luteolin, chrysoeriol, and diosmetin; flavone glycosides such as vicenin II and vitexin; and other flavone derivatives such as lactoylflavones, luteolin-7-lactate, and apigenin7-lactate can be found [2]. Among flavonoids, flavonol aglycones galangin and quercetin and their glycosides hyperoside, isoquercetin, astragalin, and rutin were analyzed. Coumarin umbelliferone and aesculin, xanthone derivative gercinone E, and naphthalene glycoside geskoidin are also found in *M. vulgare* extracts [15,22].

Other compounds determined in the herbal material of *M. vulgare* are various nitrogen-containing compounds such as choline and betonicine, amino acids and alkaloid stachydrine, polysaccharides, as well as minerals, in particular potassium salts [2,17,42]. The list of individual bioactive flavonoids in Table 4.

Marrubiin attributes include a low turnover, high stability, and low catabolism, which are core characteristics required for therapeutic compounds and nutraceuticals of economic importance. In addition, marrubiin is considered to be a potential substrate for potent active compounds, such as marrubinic acid and marrubenol [35].

The marrubiin and related diterpenoid metabolites predominantly accumulate in leaves and leaf trichomes. The central diterpenoid precursor geranylgeranyl pyrophosphate (GGPP) turns into peregrinol diphosphate, which is a bicyclic prenyl diphosphate intermediate that features a hydroxyl group at C-9, which is characteristically present in marrubiin and related metabolites [46].

The accumulation of furanic labdane diterpenes in plantlets seemingly depends on the developmental stage. No furanic labdane diterpenes were detected in plantlets during the first four to five weeks following germination. At this time, the leaves become more differentiated, and the number of trichomes on leaves is increasing. Young leaves and buds contain the highest content of furanic labdane diterpenes, which are mostly stored in peltate glandular trichomes [19].

## 4. Optimization Purification Process

In order to ensure the health benefits of herbal drugs and to avoid potential health hazards, it is necessary to provide plant material of satisfactory quality. The first step in the quality control of herbal drugs is to identify the herb. *Marrubium vulgare* can be confused with *Ballota nigra* L., *Nepeta cataria* L., *Stachy germanicus* L., *Marrubium incantum* DESR, or *Marrubium remotum* KIT [2]. For that reason, identification is of utmost importance to all subsequent tests. Herbal materials are categorized according to sensory, macroscopic, and microscopic characteristics. The other two mandatory pharmacopoeial tests are purity and content or assay with minimum 0.7% of marrubiin in dried herb [47].

The source and quality of raw materials largely depends on good agricultural practices and manufacturing processes. These are certainly essential steps for the quality assurance of herbal medicines and play the key role in guaranteeing the quality and stability of herbal preparations. All of the above-mentioned factors affect the safety and efficacy of herbal drugs [48].

Drying is used to dehydrate plants by reducing the moisture content that encourages biochemical deterioration, as well as the growth of microorganisms. The drying air temperature is a determining factor in drying kinetics. The drying rate decreases in low drying temperature. Any increment in the drying air temperature (50, 60, 70 and 80 °C) increases the energy efficiency, while any increment in the air flow rate (150 and 300 m^3^/h) shows an inverse effect. *M. vulgare* leaves dried at 80 °C and 300 m^3^/h contain relatively high amounts of total phenols in comparison with other drying temperatures [49].

Advanced control and optimization of extraction are very important in the field of chemical engineering for the development of processes that will result in high yield with minimal time and energy costs [50]. The conventional method of extraction by percolation provides 11.27% extract yield, while the microwave-assisted method of extraction gives 20.23% [14]. By using microwave-assisted solvent extraction from dry *M. vulgare* leaves and by performing analyses via a high-performance liquid chromatography ultraviolet photodiode array detection system (HPLC-UV/PAD), the best results in terms of marrubiin extraction yield were obtained by extracting samples at 120 °C with 100% ethanol for 15 min (3 × 5 min microwave cycles) [51].

Subcritical *M. vulgare* extraction can be used as a method of choice to obtain high-quality powders that preserves the constant amorphous structure after 6 months storage time. Furthermore, recoveries of both phenolic acids and flavonoids were distinctly higher when 10% maltodextrin was added as a carrier, which suggested that maltodextrin addition protects bioactive compounds from thermal degradation. This is particularly emphasized in the case of rutin content, which was 4-fold higher when a carrier was included [32]. It is reported that marrubiin, as the main active compound isolated from *M. vulgare*, can be successfully separated from complex mixtures by chitin, which is an abundant natural polymer [52].

Bouterfas et al. [53] tried to optimize the extraction conditions of phenolics, flavonoids, and condensed tannins from *M. vulgare* with water and various concentrations (20–80%) of methanol, ethanol, and acetone, as well as at various temperatures (20–60 °C) and extraction times (30–450 min). They concluded that the highest amount of phenolics (293.34 mg GAE/g DW) was obtained with 60% methanol solution at 25 °C for 180 min. The total flavonoids content was the highest (79.52 mg CE/g DW) with 80% methanol solution at 20 °C for 450 min, and condensed tannins (28.15 mg CE/g DW) with 60% acetone solutions at 50 °C and for 180 min.

Optimal conditions for obtaining essential oil are a 3% mass ratio, extraction time of 180 min, powder particles with 0.1–0.63 mm in diameter, and an extraction temperature of 120 °C. A much higher yield of essential oil is obtained from the dry plant material of *M. vulgare* than from the fresh plant. The proportion of water pretreatment is also relevant and affects the yield extraction [54]. The temperature, extraction time, particle size, ratio mass, and nature of solvent all have a positive influence on the yield of essential oil obtained by hydrodistillation and oleoresin produced by volatile solvent extraction. Miloudi et al. [55] reported that the application of a Pulsed Electric Field, as a pretreatment step for the intensification of essential oil extraction, significantly improves the extraction rate by 2–3 times.

## 5. Applications in Medicine and as Pesticides

*Marrubium vulgare* herb tea is used as a cough suppressant, and expel catarrh. ‘Materia Medica Vegetabilis’ gave directions for the preparation of *M. vulgare* decoction with honey against bronchitis and coughs [56]. The herb is used to prepare the well-known horehound candy, which, due to its pleasant taste is used to relieve cough, hoarseness, and bronchitis. It is generally recognized as safe in the USA, and it is widely used as a flavoring agent. In modern phytotherapy, various *M. vulgare* herbal medicinal products are used as an expectorant in cough associated with cold, for symptomatic treatment of mild dyspeptic complaints, such as bloating flatulence, and in temporary loss of appetite [2,57]. *M. vulgare* has a bitter value of 3.0, acting as the bitter gustative receptores (on the base of the tongue) stimulator. This effect leads to the increase in gastric and biliar secretion and the stimulation of appetite [58]. *M. vulgare* is one of the most popular herbal pectoral remedies in traditional medicine, and it is used as a bitter tonic, expectorant, and diuretic [3]. In addition to widespread use in the treatment of respiratory disorders, it has been reported that the folk use of *M. vulgare* includes treatment for jaundice, painful menstruation, and as a laxative in higher doses [59,60]. Externally, it is used for skin damage, ulcers, and wounds [23]. The importance of this plant in traditional medicine is evidenced by a number of pharmacopoeias and standard text books with monographs of *M. vulgare* [2,61]. Literature data suggest that *Marrubii herba* is traditionally used in a number of countries outside Europe [34]. Generally, its characteristic and most common use in all folk medicines is for the treatment of respiratory and gastrointestinal disorders. However, in Tunisian traditional use, this plant was also used as an hypotensive, hypoglycemic, and cardiotonic agent [62].

*M. vulgare* can be applied as herbal tea for oral use (single dose: 1–2 g of herbal material in 250 mL of boiling water, as a herbal infusion, 3 times a day; daily: dose 3–6 g), powdered herbal substance (single dose: 225–450 mg, 3 times a day; daily dose: 675–1350 mg), expressed juice (single dose: 10–20 mL, 3 times a day; daily dose: 30–60 mL) and liquid extract (single dose: 1.5–4.0 mL, 3 times a day; daily dose: 4.5–12 mL). The use in children under 12 years of age is not recommended. Safety during pregnancy and lactation has not been established [2]. However, results showed that a group of normal rats treated with the ethanol–water extract of *M. vulgare* (80:20, *v*/*v*) and pregnant rats treated with the extract exhibited a significant decrease in hematological parameters: red blood cells, hematocrit, hemoglobin, and mean corpuscular volume. The extract of *M. vulgare* caused a significant decrease on the mean implantations of fetuses and their size. As for the macroscopic and histological appearance of uterus; these data showed no change in normal treated rats. Contrary to these, the treated pregnant rats showed a severe histological change characterized by interrupted gestation, as well as the lysing of placental and embryo tissue in the uterus. All these results support the hypothesis of an abortifacient effect of *M. vulgare* [63]. Possibly, the furanic labdane diterpene marrubin and premarrubin could be responsible for these effects, since they are structural analogues with leosibiricin, an active diterpene of *Leonorus cardiaca*, which is a plant that is known for its emenagogue activity and contraindicated in pregnancy [19]. Additionally, the effects of alcoholic extract of *Marrubum vulgare* on hormonal parameters in a female rat model (receiving 500 mg/kg and 1000 mg/kg of *M. vulgare* extract for 21 days) of polycystic ovarian syndrome were studied. Luteinizing hormone (LH) hormone significantly decreased at a dose of 1000 mg/kg, and estradiol and progesterone decreased at doses of 500 mg/kg and 1000 mg/kg, while testosterone decreased at a dose of 1000 mg/kg. These findings contribute to the potential influence of examined extract on female hormones, thus questioning their safe usage in pregnancy [64]. In addition, *M. vulgare* is mentioned in certain literature as a potentially nephrotoxic plant [65]. However, there is no relevant scientific evidence to confirm these claims.

In recent years, numerous pharmacological effects of *M. vulgare* extracts have been studied. As a good antioxidant agent, *M. vulgare* proved to be very useful in treatments of cancer, diabetes mellitus, and liver diseases. In addition, many studies indicated that this plant possesses anti-inflammatory and hemostatic effects, as well as antihypertensive, sedative potential, and antimicrobial properties.

### 5.1. Antioxidant Activity

The imbalance in homeostatic processes between oxidants and antioxidants in the body, which is caused by free radicals, leads to oxidative stress. Oxidative stress is considered to be the primary cause of aging and a wide variety of human diseases, such as cancers, diabetes, neurodegenerative disorders, rheumatoid arthritis, etc. Antioxidants are substances that significantly delay, prevent, or inhibit oxidative damage to target molecules [66].

The in vitro antioxidant properties of *M. vulgare* methanol extracts were determined using DPPH (2,2-diphenyl-1-picrylhydrazyl) free radical scavenging assay and the results revealed a strong activity with the half maximal inhibitory concentration (IC_50_) value of 8.24–12.42 μg/mL [25,31]. The antioxidant activity investigated by the same method shows that *M. vulgare* essential oil exhibits IC_50_ value of 153.84 μg/mL, which is about two times higher than a synthetic antioxidant (butylated hydroxytoluene or BHT) [67]. Photochemiluminescence (PLC) assay, evaluating the antioxidant activity of the compound in the presence of superoxide anion radical, reactive oxygen species (ROS) also generated in the human body, determined the strong antioxidant effect of methanol and acetone *M. vulgare* extracts (261.41 and 272.90 μmol TE/g respectively), while the lower activity was observed when essential oil and isolated marrubin were investigated [68]. Djeridane et al. [69] determined that there is a good correlation of antioxidant potential and the content of phenolic compounds. Furthermore, acetone extracts, deodorized acetone extracts, and deodorized water extracts from *M. vulgare* leaves were tested for their antioxidant activity in rapeseed oil at 80 °C. The effect of the extracts on the edible oil stability was assessed by measuring weight gain, peroxide value, and UV absorption [70]. Acetone extracts showed better antioxidant properties than deodorized acetone extracts. According to Yousefi et al. [25], the high antioxidant activity of *M. vulgare* was associated with the presence of marrubiin, along with phenolics and flavonoids exerting a synergistic effect.

In addition, *M. vulgare* methanolic extracts showed a dose-dependent ferric reducing capacity (Ferric Reducing Antioxidant Power Assay, or FRAP) with a value equal to 50.01 μg AAE/g of extract when compared to ascorbic acid (AAE: Ascorbic Acid Equivalents) [31]. Other authors [15,28] reported that FRAP assay of ethanol–water extract of *M. vulgare* (70:30, *v*/*v*) showed IC_50_ of 64.07 mg AAE/g of dry extract and IC_50_ for the neutralization of DPPH, hydroxyl (OH), and nitroso (NO) radicals were 13.41, 63.99, and 64.86 μg/mL, respectively. Furthermore, inhibition on lipid peroxidation was examined in vitro [28], the results indicating the antiatherogenic potential of *M. vulgare* extracts.

Bouterfas et al. 2016 [71] also determined the potent activity of *M. vulgare* extracts on DPPH radical but concluded that this effect varied significantly depending on the type of the organic solvent used for extraction and sampling location.

Lastly, all these finding indicate the antiradical potential of *M. vulgare* extracts using different in vitro tests, but research should be directed toward in vivo experiments in order to elucidate their full antioxidant capacity.

### 5.2. Hepatoprotective Properties

Investigation of the hepatoprotective and therapeutic effect of ethanol–water extract (70:30, *v*/*v*) and petroleum ether extracts on CCl_4_-induced liver cell toxicity in mice showed that liver and kidney function parameters remained in the normal levels in groups treated with *M. vulgare* extracts. The administration of *M. vulgare* ethanolic extracts significantly enhanced SOD (superoxide dismutase) and CAT (catalase) activity, as well as total antioxidant capacity, with significant reduction in lipid peroxide concentration when extracts were used as protective or therapeutic agents [72]. The antihepatoxic activity of ethanol–water extracts (80:20, *v*/*v*) of *M. vulgare* in different concentrations (100, 200, 300, and 400 mg/kg) was assessed by measuring lipid profile parameters such as AST (aspartate aminotransferase), ALT (alanine aminotransferase), ALP (alkaline phosphatase), GSH (reduced gluthation), SOD, and malondialdehyde (MDA), as well as by histopathological examination of CCl_4_-induced liver damage in rats. Different concentrations of extracts showed a significant antihepatotoxic effect by reducing the levels of AST and ALT significantly, whereas the ALP level was insignificantly decreased. Regarding the antioxidant activity, these extracts exhibited a significant decrease in SOD and contents of GSH and MDA (a biomarker of membrane lipids peroxidation). These findings showed that different concentrations of *M. vulgare* extract protect liver against CCl_4_-induced hepatotoxicity, and the effect may be attributed to its antioxidant activity [73].

Methanol extracts of *M. vulgare* showed considerable antihepatotoxic effect by significantly reducing levels of AST, ALT, and LDH. However, the decrease in ALP levels was not significant. As for the antioxidant activity, *M. vulgare* extracts notably increased the GPx (glutathione peroxidases), GR (glutathione reductase), and GST (glutathione transferase) activities in rat liver tissue. In addition, it increased the GSH content and decreased the production of MDA level, adding to alleviated histopathological changes in rats’ livers treated with CCl_4_ [74].

In another animal study, marrubic acid exhibited a significant antihepatotoxic activity by reducing the elevated levels of serum enzymes such as serum glutamate oxaloacetate transaminase (SGOT) by 40.16%, serum glutamate pyruvate oxaloacetate transaminase (SGPT) by 35.06%, and alkaline phosphatase (ALP) by 30.51% [16].

The hepatoprotective potential of 12 pure compounds (marrubiin, premarrubiin, vulgarin, luteol, vulgarol, apigenin-7-glucronide, vitexin, apigenin, chryseriol, stachydrine, acetoside, and 1-caffeory-L-malic acid) characteristic for *M. vulgare* were tested on CCl_4_-induced acute liver injury in rats through the in silico method. All compounds showed expected and similar bioactivity, especially when it comes to liver-associated enzymes inhibition [17].

### 5.3. Antiproliferative Activity

*M. vulgare* are often used traditionally in cancer treatment [75,76], but the exact mechanisms of action and scientific validity of their usage are yet to be discovered.

Zarai et al. [77] reported the ability of *M. vulgare* essential oil to inhibit the proliferation of cervical cancer (HeLa) cell lines with an IC_50_ value of 0.258 μg/mL. *M. vulgare* ethanol–water extracts (70:30, *v*/*v*) reduced the viability of melanoma (B16) and glioma (U251) in a dose-dependent manner. By demonstrating the ability of *M. vulgare* extracts to inhibit proliferation, induce apoptosis, and cytoprotective autophagy, the results suggested that this plant could be a good candidate for anti-melanoma and anti-glioma therapy [22]. In addition, the methanolic extract of *M. vulgare* was evaluated for its in vitro cytotoxic activity by measuring the percentage of viable cells’ glioblastoma multiforme cell lines (U87, LN229 and T98G) using a luminescence system. After evaluating the cytotoxicity impact that *M. vulgare* has on U87 (IC_50_: 270.3 μM), LN229 (IC_50_: 343 μM), and T98G (IC_50_: 336.6 μM) from glioblastoma multiforme cell lines, it was concluded that it is the most efficient in two cell lines, U87 (69.9%) and LN229 (71%) [30].

The in vitro anticancer activity of *M. vulgare* ethanol–water extracts (90:10, *v*/*v*) and six pure compounds (acacetin, acacetin-7-rhamnoseide, apigenin, diosmetin, diosmetin-7-glucoside, and luteolin-7-rhamnoside) were also tested against Ehrlich tumor cell lines, human tumor cell lines U251 (brain tumor) and MCF7 (breast cell lines) [43]. Alcoholic extracts, acacetin, apigenin, and acacetin-7-rhamnoside showed high anticancer activity against breast carcinoma, whereas all tested compounds had anticancer activity against Ehrlich tumor cell lines. Another study [45] showed that labdanein (methoxylated flavone) from *M. vulgare* displayed a moderate effect on human myeloid leukemia (K562) and imatinib-resistant human myeloid leukemia (K562R) cells, as well as on human B cell precursor leukemia cell lines (697). The authors of this study suggested that these results provide a common natural source for the hemi-synthesis of future ladanein-derived flavones and the study of their antileukemic activity. Tlili et al. [78] examine the effect of *M. vulgare* and other Tunisian plant extracts on leukemia and colon cancer cell lines (K-562 and CaCo-2, respectively). Subsequently, the anti-inflammatory activity was assessed, and the results showed that white horehound possesses the highest activity in the group of analyzed plants.

Kozyra et al. [79] investigated the potential anticancer activity of methanolic extracts phenolic acid (PhA) fractions of *M. vulgare* against a human melanoma cancer cell line (A375) and normal human skin fibroblasts (BJ) using a 3-(4,5-dimethylthiazol-2-yl)-2,5-diphenyl-tetrazolium bromide test, cell cycle analysis, and real-time monitoring of cell viability. Surprisingly, examined fractions demonstrated a low total phenolic content and did not show significant antioxidative properties, but the nonhydrolyzed PhA fraction exhibited cytotoxic activity against a human melanoma cancer cell line, without affecting normal fibroblasts. Both acidic and alkaline hydrolysis abolished this activity, indicating that the esterified forms of phenolic compounds exhibit the observed cytotoxic effects. Since *M. vulgare* is abundant in phenylpropanoid (cinnamic) esters and phenylethanoid glycosides, further investigation of these compounds may provide an insight into the exact anticancer mechanism of action and use as a model for cancer treatment drug development.

Moreover, Shawky [80] used the network pharmacology approach to identify the main active constituents of North African plants against cancer molecular targets and to explore their therapeutic mechanism. *M. vulgare* possessed the largest number of plant–constituent–target gene interactions indicating cell cycle arrest and apoptosis in addition to the inhibition of cellular proliferation as possible mechanisms and marked this plant as a potential source for the supportive treatment of cancer.

### 5.4. Anti-Inflammatory Activity

Investigation of the anti-inflammatory effect of the methanolic extracts of *M. vulgare* on isoproterenol-induced myocardial infarction in a rat model showed that serum creatinine kinase-MB was subsided by 52.2–69.0% (depending on the dose of *M. vulgare* extract). In addition, the treatment with extracts significantly reduced myocardial myeloperoxidase activity in myocardial infarction [81,82]. Levels of tumor necrosis factor alpha (TNF-α) also declined considerably in the serums of rats with myocardial infarction. In addition, peripheral neutrophil count was significantly lowered by all doses of the extract. Interstitial fibrosis was significantly attenuated by treatment with *M. vulgare* compared to control. The results of this study demonstrated that *M. vulgare* extracts have strong protective effects against isoproterenol-induced myocardial infarction, and it seems possible that this protection is a result of its anti-inflammatory properties. Furthermore, the 11-oxomarrubiin, vulgarcoside A, and 3-hydroxyapigenin-4′-*O*-(6′′-*O*-p-coumaroyl)-β-d-glucopyranoside from the methanolic extract of *M. vulgare* exhibited moderate to low levels of inhibition on NO production, while vulgarcoside A also showed a moderate inhibition effect on pro-inflammatory cytokinine TNF-α [20]. Glycosidic phenylpropanoid esters from *M. vulgare* showed inhibitory activity toward the cyclooxygenase (COX) enzyme, which plays a key role in the transformation of arachidonic acid into pro-inflammatory prostaglandins and is associated with inflammation [40].

The assessment of anti-inflammatory activity showed that the oral administration of methanolic extract of *M. vulgare* at a dose of 200 mg/kg in rats treated with carrageenin causes a significant decrease (87.30%) of inflammation compared to standard positive control (diclofenac), which showed 85.52% protection in this test [31]. In a model of microvascular leakage in mice ears, the analysis shows that marrubiin from *M. vulgare* exhibits significant and dose-related antioedematogenic effects. The treatment of mice with marrubiin caused a dose-dependent inhibition of carregeenan, bradykinin, and histamine-induced extravasation of Evans blue in mice ears, with maximal inhibitions of 63.0%, 70.0%, and 73.7%, respectively. The other phlogistic agonists, such as prostaglandin E2, caused an inhibition of less than 50%. In addition, marrubiin significantly inhibited the ovalbumin-induced allergic edema in actively sensitized animals. These results demonstrate that the systemic administration of marrubiin exerts a non-specific inhibitory effect on pro-inflammatory agent-induced microvascular extravasation of Evans blue in mice ears [36]. The evaluation of anti-inflammatory activities against inflammation induced by carrageenen and prostaglandin E2 and analgesic activity in the *p*-benzoquinone-induced abdominal constriction test showed that methanolic extracts of *M. vulgare* have a similar effect as reference drugs indomethacin and acetylsalicylic acid [59].

An in vitro investigation of the anti-inflammatory effect that six compounds from *M. vulgare* (luteolin-7-*O*-β-glucopyranoside, apigenin-7-*O*-β-glucopyranoside, oleanolic acid, β-sitosterol, luteolin-7-*O*-rutinoside, and rosmarinic acid) have on COX showed that these compounds inhibited the formation of hormones, such as prostaglandins and peroxasalandine, which contribute to the production of inflammatory intermediators [18].

### 5.5. Immunomodulatory Activity

The immunomodulatory effect of different concentrations of *M. vulgare* aqueous extracts (100, 500, and 1000 mg/kg body weight) was evaluated as a cure agent in mice previously infected with *Salmonella typhimurium* and as a protective agent in mice infested with *S. typhimurium* [83]. According to these results, the lowest concentration of *M. vulgare* (100 mg/kg body weight) showed high immunomodulatory effect in the level of double positive T cells, interleukin (17AIL-17) and interferon-gamma (IFN-γ).

### 5.6. Sedative Activity

The potential to reduce morphine withdrawal signs in animals had been studied using aqueous and ethanolic extracts of *M. vulgare* in different doses (0.1, 0.5, 1.5, and 2.5 g/kg). It was concluded that all doses reduced the physical activity of mice. They also induced muscle relaxation [84].

### 5.7. Antidiabetic Activity

*Marrubium vulgare* has an ethnomedical record as an antidiabetic agent [85,86]. Certain efforts have been made in order to obtain scientific evidence supporting its traditional use in diabetes mellitus control [87]. Hellal et al. [88] performed in vitro screening for the α-glucosidase inhibitory activity of six Algerian traditional medicinal plant extracts where *M. vulgare* 80% ethanol extracts exerted a moderate effect (IC_50_ = 12.66 μg/mL). Moreover, a series of in vivo experiments were carried out on alloxan-induced diabetes during which the experimental animals were treated twice a day with aqueous extract of *M. vulgare* for 15 days [27]. Oral administration of 200 and 300 mg/kg body weight of *M. vulgare* aqueous extract induced significant dose-dependent antidiabetic and antihyperlipidemic effects. A dose of 100 mg/kg reduced blood glucose by 50%, whereas doses of 200 and 300 mg/kg showed more than 60% reduction of the same parameter. A decrease of total lipids, triglycerides, and cholesterol levels were observed in animals treated with *M. vulgare* when compared to the diabetic control group. Glibenclamide was used as a reference and showed similar effects, and the authors hypothesized that flavonoids and verbascoside derivatives present in examine extract caused the observed effects [27].

Chakir et al. [89] showed that the oral administration of *M. vulgare* methanolic extracts resulted in a significant lowering of blood glucose level, serum urea, uric acid, and creatinine, as well as a correction of lipid profiles when compared to streptozotocin-induced diabetic rats. These methanolic extracts significantly increased the glucose uptake of liver and skeletal muscles. Contrary to this, they reduced the glucose absorption in the everted rat jejunum. These results suggest that the effect of *M. vulgare* extract may be due to extrapancreatic mechanisms. This antidiabetic effect is an outcome of the modulation of glycogen synthesis and the inhibition of intestinal glucose absorption.

Another study [90] showed the effect of different *M. vulgare* extracts (methanol, water, and buthanol) on autoimmune diabetes mellitus induced by cyclosporine and streptozotocin. When compared to diabetic mice, the animals from the group treated with extracts of *M. vulgare* showed a decrease in blood glucose levels, pancreatic levels of interferon gamma (IFN-γ), and NO. *M. vulgare* extracts also caused a significant decrease in total cholesterol, low-density lipoprotein (LDL) cholesterol, very-low-density lipoprotein (VLDL) cholesterol, and triglycerides. Additionally, the serum insulin levels were significantly increased after treatment with *M. vulgare.*

In addition, Alkofahi et al. [91] screened 21 plants grown in Jordan for their antihyperglycemic activity on Sprague–Dawley rats at 1 g/kg where *M. vulgare* extract showed a neutral effect on blood glucose levels.

The incidence of atherosclerosis and cardiovascular diseases increases in diabetes mellitus patients. Therefore, the effects of *M. vulgare* on the contractile reactivity of isolated aorta were analyzed in an experimental model of streptozotocin-induced diabetic rats after two months of oral administration of *M. vulgare* [92]. The results showed that serum glucose levels increased significantly in diabetic rats, while this increase was not observed in animals treated with *M. vulgare*. In addition, *M. vulgare*-treated rats showed a lower concentration of KCl and noradrenaline in comparison to the control group. Based on these results, it was concluded that the oral administration of *M. vulgare* during 2 months could attenuate the contractile responsiveness of the vascular system, which may prevent the development of hypertension in diabetic rats.

Although there is an evident antihyperglycemic effect of *M. vulgare* extract on induced diabetes in experimental animals, the duration of these studies were short, and also another model of type 2 diabetes studies would be interesting to elucidate the exact mechanisms of action of this plant. In addition, there has been one clinical trial that contradicted findings from animal studies. Namely, aqueous extract (infusion of dried *M. vulgare* leaves) was tested to evaluate a clinical effect in 22 type 2 diabetic patients that had poor response to conventional treatment. The results showed that infusion reduced glucose levels by only 0.64%, cholesterol by 4.16%, and triglycerides by 5.78% [93]. Finally, novel clinical trials are essential in order to confirm antidiabetic activity in humans as well as determine the right therapeutic protocol, potential adverse effects, and precautions.

### 5.8. Antihypertensive Activity

The water extract of *M. vulgare* is widely used as an antihypertensive treatment in folk medicine. Crude extracts of the aerial parts of *M. vulgare* show a potent in vitro inhibition of KCl-induced contraction of rat aorta. Bioguided fractionations, spectroscopic analysis, and chemical derivatization revealed furanic labdane diterpenes, marrubenol, and marrubiin as the most active compounds [33,37]. By analyzing the effects that 10-week-long treatment with amlodipine and *M. vulgare* water extract had on the systolic blood pressure, cardiovascular remodeling, and vascular relaxation in spontaneously hypertensive rats, it was observed that treatment with *M. vulgare* produced a decrease in systolic blood pressure. In addition, it had significant antihypertrophic effect in aorta and it improved acetylcholine (ACh)-induced relaxation of mesenteric artery. These results demonstrated that in addition to its antihypertensive effect, *M. vulgare* water extract improved the impaired endothelial function in spontaneously hypertensive rats [94].

### 5.9. Hypolipemic Activity

One study evaluated the hypocholesterolemic and hypotriglyceridemic activities of four *M. vulgare* herb extracts (petroleum ether, chloroform, ethyl acetate, and methanol) using Triton WR-1339-induced hyperlipidemia in mice. Extracts were applied using 0.1 and 0.25 LD_50_ concentrations. After 7 h and 24 h of treatment, the intragastric administration of all extracts caused a significant decrease of plasma total cholesterol; LDL cholesterol and triglyceride levels were also significantly lowered by all extracts. Additionally, more polar extracts (methanol and ethyl acetate) showed a significant ameliorative action on elevated atherogenic index (AI) and LDL/HDL-C ratios [95]. These findings coincide with recorded decreases in the total cholesterol and LDL cholesterol levels during antidiabetic studies, which was probably due to a stimulation of the insulin secretion [86].

Moreover, since metabolic syndrome as a cluster of conditions includes increased blood pressure, high blood sugar, and abnormal cholesterol or triglyceride levels, all of which are influenced by *Marrbium vulgare* herb extract, it would be useful to evaluate its possible usage as a part of nutrcaceuticals approach to this indication [96].

### 5.10. Gastroprotective Activity

In the model of ethanol-induced ulcers, a significant reduction in all analyzed parameters was observed when *M. vulgare* extract was applied [34]. The curative ratios were 49.31–74.31% for the groups treated with 50 and 100 mg/kg of *M. vulgare* extract. For indomethacin-induced ulcers, the percentages of ulcer inhibition were 50.32%, 66.24%, and 82.17% for the groups treated with 25, 50, and 100 mg/kg of *M. vulgare*. In both models, the marrubiin (25 mg/kg) produced a significant reduction in all parameters compared to the control group. There was also a significant increase in pH and mucus production in groups treated with *M. vulgare* extracts and marrubiin. The results demonstrated that the gastroprotective effect induced by these extracts and marrubiin is related to the activity of NO and endogenous sulfhydryls, which are important gastroprotective factors.

### 5.11. Antimicrobial Activity

The antibacterial potential of different *M. vulgare* extracts (ethyl acetate, diethyl ether, and 1-butanol) was tested on four strains of bacteria: *Staphylococcus aureus*, *Escherichia coli*, *Proteus vulgaris*, and *Pseudomonas aeruginosa*. The antibacterial activity of different fractions performed on solid agar medium showed little or no effect. This implies that the antibacterial activity proved with crude extract of *M. vulgare* was likely induced by synergistic action in chemical ingredients present in the extracts [97]. The methanolic extract of *M. vulgare* showed a significant antimicrobial activity against *Escherichia coli*, *Bacillus subtilis*, *Staphylococcus aureus*, *S. epidermidis*, *Pseudomonas aeruginosa*, *Proteus vulgaris*, and *Candida albicans* [59].

Moreover, one study was undertaken to determine the antifungal activity of flavonoids (flavans and flavanols) extracted from the leaves of *M. vulgare* against two fungal strains; *Aspergillus niger* ATCC 16,404 and *Candida albicans* ATCC 10,231 using the solid medium diffusion method. The minimum inhibitory concentrations (MICs) obtained range between 6.25 and 100 μg/mL and led to experiencing strong antifungal inhibition, which often exceeded the effect of marketed antifungals (amphotericin, fluconazole, terbinafine, and econazole nitrate) that marked *M. vulgare* flavonoids as potentially powerful antifungal agents [98]. In addition, Rezgui et al. [68] concluded that *M. vulgare* and marrubiin can be used as antifungal agents for the treatment of skin dermatophyte infections. They examined the effect of acetone and methanol extracts, essential oil, and marrubin (all in two doses: 20 and 100 μg/mL) against the dermatophytes fungi *Microsporum gypseum*, *Microsporum canis*, *Arthroderma cajetani*, *Trichophyton mentagrophytes*, *Trichophyton tonsurans*, *Epidermophyton floccosum*, and against two fungi strains (*Botrytis cinerea*, *Pythium ultimum*).

*M. vulgare* essential oil has a significant effect on microorganisms, especially Gram+ bacteria with inhibition zones and MIC values in the range of 6.6–25.2 mm and 1120–2600 μg/mL, respectively, whereas Gram– bacteria exhibited higher resistance. When it comes to antifungal activity, among the four strains tested, *Botrytis cinerea* exhibited the strongest response to M. vulgare essential oil, with inhibition zones of 12.6 mm. However, *Fusarium solani*, *Penicillium digitatum*, and *Aspergillus niger* were less sensitive to this essential oil [77].

### 5.12. Wound Healing (Hemostatic)

The study of use of methanolic extract of *M. vulgare* in wound reparation demonstrated that this extract, which was rich in poliphenolic compounds (flavonoids and several phenylethanoid glycosides) and marrubiin (6.62%), showed antioxidant and wound-healing properties by promoting cell migration and the proliferation of fibrosis [23]. The assessment of hemostatic activity through the plasma recalcification method confirmed the surprising dose-dependent anticoagulant effect of aqueous extract of *M. vulgare* [26]. A positive linear correlation between the studied parameters, content of condensed tannins, and hemostatic activity was used to highlight the potent vasoconstriction property of these compounds.

### 5.13. Antiviral Activity

Acute and recurrent herpes simplex virus type 1 (HSV-1) infections remain an important problem due to the emergence of acyclovir resistant virus. As a result of that, the search for novel antiviral bioactive compounds from plants has intensified in recent years. The antiviral activity of methanol, hexane, and chloroform extract of *M. vulgare* showed antiviral activity with selectivity indices of 3.11, 2.8, and 1.28, respectively. This study revealed that the hexane fraction disrupts the early steps of cyclic replication, including HSV-1 attachment, in a dose-dependent manner [99].

### 5.14. Antiparasitic Activity

*Toxoplasma gondii* is an intracellular parasite that causes many symptoms, such as encephalitis and congenital disorders. Using MTT (3-(4,5-dimethylthiazol-2-yl)-2–5-diphenyltetrazolium bromide) cell-proliferation assay *in vitro*, it was concluded that *M. vulgare* could be very useful against this parasite [100].

### 5.15. Antiprotozoal Activity

*M. vulgare* essential oil and extracts (n-hexane, ethyl acetate, and methanol) have potent antiprotozoal activity against *Trichomonas vaginalis*. After 48 h of exposure, the essential oil with a minimal inhibitory concentration value of 291 μg/mL showed the highest antiprotozoal activity, followed by ethyl acetate (541 μg/mL), methanol (1000 μg/mL), and n-hexane (1500 μg/mL). According to the findings of this study, the compounds of *M. vulgare* have a significant effect on *T. vaginalis* [101].

### 5.16. Antiplasmodial Activity

The antiplasmodial activity of ethanolic extract of *M. vulgare* was evaluated in mice infected with chlorquine-sensitive *Plasmodium berghei-berghei* using curative, suppressive, and prophylactic experimental animal models. Preliminary phytochemical screening and intraperitoneal LD_50_ (50% lethal dose) estimation of the extract were carried out. In all doses tested, the extract produced significant curative and suppressive effects with minimal prophylactic effect. The extract also significantly prolonged the survival time of treated mice (up to 22 d), compared to the negative control group (11 d). The results of this study suggest that the ethanol extract of *M. vulgare* possesses curative and suppressive antiplasmodial activity in mice at all tested doses [102].

### 5.17. Veterinary Medicine

The anthelmintic activities of ethanolic and aqueous extract of *M. vulgare* were evaluated using the egg hatch assay and larval mortality assay. After a 24 h exposure period at a concentration of 50 mg/mL, the mortality rate was 45.8% for the aqueous extract and 51% for the ethanolic extract. These findings showed that *M. vulgare* extracts have potential anthelmintic effect on eggs and larvae of bovine strongyles parasites in vitro [103].

Bovine mastitis is the most serious diary problem in terms of economic losses to the dairy industry. With new generations of virulence and resistant bacteria, finding alternative treatments with medicinal plants to control these pathogenic strains is very popular. Among others, the ethanol extract of *M. vulgare* shows good antibacterial activity against methicillin-resistant staphylococci and multi-resistant *Escherichia coli* strains isolated from animals with mastitis manifestation [104].

### 5.18. Use as Natural Pesticides

In Spain, *M. vulgare* has been popularly used on chicken farms to prevent lice and frequent scratching of animals, which intensified its planting on farms [12]. Moreover, a plant extract of *M. vulgare* was tested against fourth instar larvae of the mosquito *Culex pipiens*. The obtained results indicated a sensitivity of *C. pipiens* larvae that was even higher when the exposure time of larvae to the insecticide is extended. The greatest mortality rate (94%) was achieved with 900 mg/L and a 72 h exposure to *M. vulgare* extract, whereas a 59% mortality rate was achieved with 900 mg/L and a 72 h exposure period. These results may provide an opportunity to develop alternatives to environmentally hazardous chemicals using some readily available, affordable plants which are mostly harmless to different living organisms [105]. Furthermore, the volatile oil of *M. vulgare* has a remarkable toxicity on the snails of both *Schistosoma mansoni* and *S. haematobium* species [106]. In addition, *M. vulgare* leaf extract and rizosphere soil extract significantly influence the seed germination and seedling growth of *Sinapis arvensis* and *Latuca sativa* in laboratory conditions. However, the allelopatic effects depend on target species. These extracts can be used as an important source of natural herbicides to control weeds in crop fields [29].

## 6. Conclusions

*M. vulgare* produce structurally highly diverse groups of secondary metabolites, thus representing the valuable source of bioactive compounds and preparations with health-promoting effects: antioxidant, hepatoprotective, antiproliferative, anti-inflammatory, antidiabetic, and antimicrobial being the most investigated. Although these effects of *M. vulgare* extracts, essential oil, marrubiin, flavan and flavonol type of flavonoids, and phenylethyl esters were studied, the pharmacokinetics of these compounds and the concentrations of extracts, essential oil, and isolated compounds needed for the pharmacological effect have yet to be established. White horehound is a part of traditional medicine systems worldwide; it is generally recognized as safe, but well designed clinical trials are needed in order to shift from traditional to a well-established use of *M. vulgare* herb preparations for the prevention and treatment of various ailments.

## Figures and Tables

**Table 1 molecules-25-02898-t001:** Quantification of phenolic and flavonoid content in *M. vulgare*. (GAE—gallic acid equivalents, QE—quercetin equivalents, CE—catechin equivalents, RUE—Rutin equivalent, na-not avalilable).

Extraction	Phenolic Content	Flavonoid Content	References
Aqueous extract	175.00 mg GAE/100 g dry material	23.86 mg QE/100 g dry material	[26]
Aqueous infusion	na	5.08 mg/100 mg dry material	[27]
Ethanol–water extract (70:30, *v*/*v*)	93.42 mg of GAE	23.25 mg of RUE/g of extract	[14]
Ethanol–water extract (70:30, *v*/*v*)	59.87 mg GAE/g dry extract	14.47 mg QE/g dry extract	[28]
Methanol extract (dry)	44.89 mg GAE/g dry extract	24.60 mg QE/g dry extract	[29]
Methanol extract	48.97–195.00 mg GAE/g extract	12.05–93.12 mg QE/100 g extract	[25,30,31]
Microwave assisted extract	61.44 mg of GAE/g of extract	37.70 mg of RUE/g of extract	[14]
Subcritical water extract	72.98–85.20 mg GAE/g extract	26.59–31.37 mg CE/g extract	[32]

**Table 2 molecules-25-02898-t002:** List of bioactive non-volatile terpenoid compounds from *M. vulgare* extracts.

Compound	References	Structure
***Monoterpene derivatives***
Marrubic acid	[16]	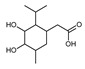
Sacranoside A	[15]	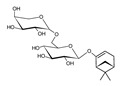
***Sesquiterpene lactone***
Vulgarin	[17]	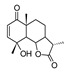
***Diterpenoids***
Marrubiin	[17,19,23,33,34,35,36]	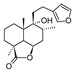
12(*S*)-hydroxymarrubiin	[24]	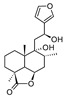
11-Oxomarrubiin	[20]	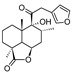
3-Deoxo-15(*S*)-methoxyvelutine	[24]	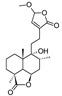
Marrubenol	[15,33,37]	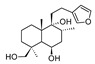
Premarrubiin	[15,17]	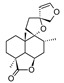
Marruliba-acetal	[15]	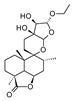
Cyllenin A	[21]	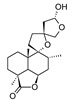
Polyodonine	[20]	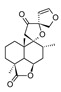
Preleosibirin	[15]	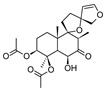
Peregrinol	[21]	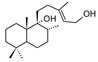
Peregrinin	[24]	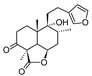
Dihydroperegrinin	[24]	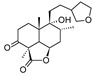
Vulgarol	[17]	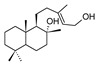
Vulgarcoside A	[20]	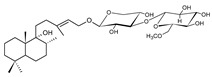
Deacetylvitexilactone	[23]	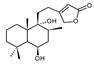
Carnosol	[22]	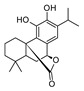
Deacetylforskolin	[15]	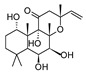
***Phytosterol***
β-sitosterol	[18]	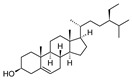
***Triterpenoids***
Lupeol	[17]	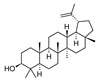
Oleanolic acid	[18]	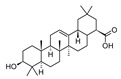

**Table 3 molecules-25-02898-t003:** List of bioactive phenolic acids and esters from *M. vulgare* extracts.

Compound	References	Structure
***Coumarins***
Umbelliferone	[22]	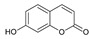
Aesculin	[22]	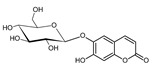
***Phenolic acids***
Gallic acid	[22]	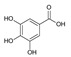
Gentisic acid	[22]	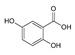
*p*-Hydroxybenzoic acid	[22]	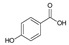
Protocatechuic acid	[22]	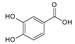
Syringic acid	[39]	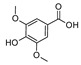
***Phenylpropanoid (cinnamic) acids***
*trans*-Cinnamic acid	[39]	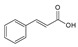
Caffeic acid	[15,22,32,39]	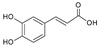
Ferulic acid	[22,32,39]	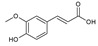
*o*-Coumaric acid	[39]	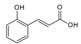
*p*-Coumaric acid	[22,32,39]	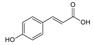
Sinapic acid	[39]	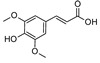
***Phenylpropanoid (cinnamic) ester***
Caffeoylmalic acid	[17,40]	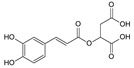
Rosmarinic acid	[18,22,39]	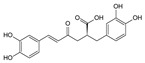
Chlorogenic acid	[22,27]	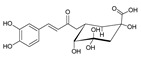
***Phenylethanoid glycosides***
Acteoside (1)	[15,17,23,27,40,41]	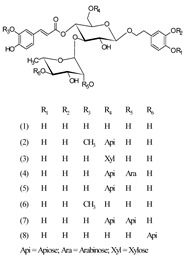
Alyssonoside (2)	[15,27]
Arenarioside (3)	[40]
Ballotetroside (4)	[27,40]
Forsythoside B (5)	[15,23,40,41]
Leucosceptoside A (6)	[15]
Marruboside (7)	[15,23,40]
Samioside (8)	[23]

**Table 4 molecules-25-02898-t004:** List of bioactive flavonoids from *M. vulgare* extracts.

Compound	References	Structure
***Flavanone***		
Naringenin (1)	[22]	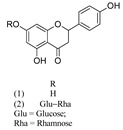
Naringin (2)	[22]	
***Flavone***		
Acacetin (1)	[43]	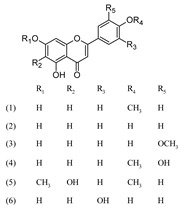
Apigenin (2)	[15,17,20,22,39,43,44]	
Chrysoeriol (3)	[17,44]	
Diosmetin (4)	[43]	
Ladanein (5)	[27,41,45]	
Luteolin (6)	[15,22,44]	
***Flavone derivatives***		
Acacetin 7-*O*-rhamnoside (1)	[43]	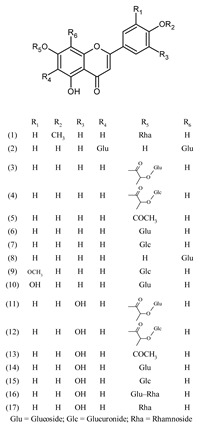
Apigenin 6,8-di-*C*-glucoside (2) (Vicenin II)	[27,44]	
Apigenin 7-[2-glucosyllactate] (3)	[44]	
Apigenin 7-[2-glucuronosyllactate] (4)	[44]	
Apigenin 7-acetate (5)	[44]	
Apigenin 7-*O*-glucoside (6)	[15,18,27,44]	
Apigenin 7-*O*-glucuronide (7)	[17,27]	
Apigenin 8-C-glucoside (8) (Vitexin)	[17,44]	
Chrysoeriol 7-*O*-glucuronide (9)	[27]	
Diosmetin-7-O-glucoside (10)	[43]	
Luteolin 7-[2-glucosyllactate] (11)	[44]	
Luteolin 7-[2-glucuronosyllactate] (12)	[44]	
Luteolin 7-acetate (13)	[44]	
Luteolin 7-*O*-glucoside (14)	[15,18,27,39,41,44]	
Luteolin 7-*O*-glucuronide (15)	[27,41]	
Luteolin 7-*O*-rutinoside (16)	[18]	
Luteolin 7-rhamnoside (17)	[43]	
3-hydroxyapigenin 4′-*O*-(6′′-*O*-*p*-coumaroyl)-glucoside (1)	[20]	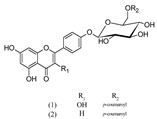
Apigenin 4′-*O*-(6′′-*O*-*p*-coumaroyl)-glucoside (2)	[20]	
Apigenin 7-*O*-(4′′-*p*-coumaroyl)-glucoside (1) (Terniflorin)	[15]	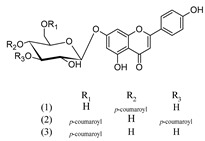
Apigenin 7-*O*-(3′′, 6′′-di-*p*-coumaroyl)-glucoside (2) (Anisofolin A)	[20]	
Apigenin 7-*O*-(6′′-*p*-coumaroyl)-glucoside (3)	[44]	
***Flavonol***		
Galangin (1)	[22]	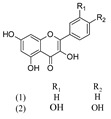
Quercetin (2)	[22]	
***Flavonol derivatives***		
Quercetin 3-*O*-galactoside (1) (Hyperoside)	[32,39]	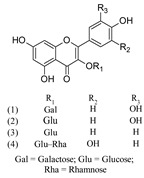
Quercetin 3-*O*-glucoside (2) (Isoquercetin)	[39]	
Kaempferol 3-*O*-glucoside (3) (Astragalin)	[39]	
Quercetin 3-*O*-rutinoside (4) (Rutin)	[22,32]	

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
