# Peer review of "Marrubium vulgare L.: A Phytochemical and Pharmacological Overview"

_molecules, 2020, doi:10.3390/molecules25122898_

Round 1

Reviewer 1 Report

The manuscript (molecules-836764), titled “Marrubium vulgare – chemistry and bioactivity” presents the main activities attributed to the species as well as the secondary metabolites isolated from this medicinal plant. It is an interesting descriptive work but, in my opinion, should be revised before it could be published.

  1. Point 5 should appear before the points 3 and 4. Because it is explaining why the species is a medicinal plant. Points 6, 7, 8, and 9 are also related to the species use. Probably they should all be included in point 5, changing the title for traditional uses, which will consist of the applications in medicine and as pesticides.
  2. Points 2 and 10 should be joined, or at least do not be so separated. Simultaneously point 11 seems to be more related to point 2.
  3. The points presented seem to be out of order as if different authors did them, and no one connected every information. In this way, it is difficult to understand the significance of the species. A careful organization is needed.
  4. Table 1 gives no new information, flavonoids are phenolic compounds, and the presence of these classes is frequent in all plants. The authors do not establish a relation between these contents and compounds isolated or biological activities. So the information is of no importance.
  5. Again there is no relationship between the compounds isolated and the bioactivities found or the traditional applications. Some efforts should be made to establish those relationships that help the readers understand the value of the plant. Some conclusions should be added at the end of the manuscript.
  6. Table 2 should be reorganized. By families, such as flavonoids, terpenoids, etc.…And, inside each family could appear in the classes. For example, the flavonoids subdivided into flavones, flavanones, etc.… The way it is presented is confusion, and the readers will need to scroll several times to find valuable information. It would be interesting to relate these bioactive compounds with the activities described in point 4.
  7. Some conclusions are needed; point out the plant value as a source of bioactive compounds and health-promoting effects. But also indicating what else should be done. Are there clinical trials? Point 7 is too vague. The information about in vivo studies is dispersed in point 4, and it should be more precise. In vivo studies, clinical trials and cytotoxic studies are essential and should be highlighted.

Author Response

Dear reviewer,
we thank you for highly constructive suggestions which have significantly contributed to quality of this paper.
Having in mind that almost all reviewers advised rearrangement of subtitles, the new subtitles are as follows:
  1. Introduction
  2. Botanical features
  3. Phytochemical composition
  4. Optimization purification process
  5. Applications in medicine and as pesticides
5.1. Antioxidant activity
5.2. Antiproliferative activity
5.3. Gastroprotective activity
5.4. Antidiabetic activity
5.5. Immunomodulatory activity
5.6. Hepatoprotective properties
5.7. Antiinflamatory activity
5.8. Wound healing (haemostatic)
5.9. Antihypertensive activity
5.10. Sedative activity
5.11. Antimicrobial activity
5.12. Antiviral activity
5.13. Antiparasitic activity
5.14. Antiprotozoal activity
5.15. Antiplasmodial activity
5.16. Veterinary medicine
5.17. Use as natural pesticides
  1. Conclusion
Table 1 was kept with minor changes, whereas Table 2 was modified, divided into 3 new tables, according to families.

Sincerely,

Authors

Reviewer 2 Report

Reviewers' comments: molecules-836764

Reviewer #:

The research article led by Milica et al. entitled “Marrubium Vulgare-Chemistry and bioactivity”.

It was documented about the potentiality of Marrubium Vulgare (M. Vulgare) as a bioactive compound which was isolated by three optimal conditions such as water extract, Ethanol-water extract or methanol-water extract along with isolated and purified single compounds by categorized as followings: Alkaloid, Diterpenoids, coumarin, flavanone, flavone, flavonol, monoterpene, Naphthalene glycoside, Phenolic acids, Phenylethanoid glycosides, phenylpropanoid ester, phytosterol, a sesquiterpene lactone, triterpenoids, Xanthine, etc.

Those bioactive compounds are worth to challenge to reveal underlying mechanisms using the specific animal models or targeted molecules in detail. I am not sure if authors cover the title with concentrated the single compound or M Vulgare, in particular Phenolic and flavonoid in general. I am concerned one point There are no specificity and selective difference Phenolic and flavonoid in M. Vulgare and different herb or Plants. Even though the authors indicated the issue of dosage and safety with regard to the extract, not single compound among any Phenolic and flavonoids in M Vulgare.  It looks benefits with an integrated summary of potential phytochemical, M. Vulgare, in the field of pharmaceutical, natural product and chemistry, and medicinal chemistry, etc. Within Medicinal properties (line 93), the authors introduced several potentialities of M. Vulgare extract by citation several experimental pieces of evidence. However, there is a lack of single compound like the impact of Marruiin on various bioactivities regarding 15 fields as indicated in the context.

The length of the article is adequate; well developed their objective sequentially following by the introduction, phytochemical morphology with regard to potential SAR (structure-activity relation) with pieces of evidence in the references cited relevantly. However, I found there is some degree of incompleteness, a kind of clarity in data difference regarding phenolic acids and flavonoid versus Marrubiin (Diterpenoids) by which they may claim as highlighted significance in this type of novel phytochemical from M. Vulgare covering various multi-functionality in cellular health and immune. The shortage of this review did not describe what molecular target could be counteracted by Marrubiin or phenolic acids and flavonoids in the disease pattern or model.

Minor:

In the main title, we recommend potential topic like as  Marrubium vulgare: structure diversity with multi-functionality in the photochemistry

Line 77, please add RUE mean this abbreviation as a footnote; I suggest NA, Not available, instead of du.in the table 1 at page 2

Please check reference 74 on page 14, I can’t see that.

On page 14, I’d like to discuss to subtitle 11. Postharvest processing. Actually, If my understanding was correct, the authors focused on the optimization of phenolic acids and flavonoids from  M. Vulgare. Hence, I suggest that the Optimization purification process is better than the current subtitle.

Author Response

Dear reviewer,

Thank you for the highly constructive comments. We made the effort to modify the paper.

After significant changes, the subtitle 3. Phytochemical composition is significantly improved.

Sincerely,

Autors

Reviewer 3 Report

Present review is aimed at comprehensively describing the botanical features, phytochemicals and pharmacological properties of Marrubium vulgare along with to highlight the interest in this medicinal plant as a source of herbal remedies. This interest is supported by the traditional use of M. vulgare preparations in medicine as stated by the EMA commission. Although it could be of interest an update about the current knowledge on this plant as reported in the present review, there are some points that requires further improvements.

  • The aim of the study and the novelty of this work should be better highlighted.
  • Methodology applied to select the reviewed literature should be defined in the introduction.
  • An update of the clinical evidence that supports the medical use of vulgare could markedly increase the scientific impact of this review.
  • A “Conclusions” paragraph should be included in order to highlight specific points and further developments of this plants.
  • Some typing and formatting errors should be checked and corrected.

Other points to be considered:

  • Title should be improved by better describing the content of the review.
  • The correct binomial names Marrubium vulgare should be reported.
  • Abstract (lines 9-12): the wording “with impressive bioactive potential” and “very useful in treatments of cancer, diabetes mellitus and liver diseases” are too emphasized. There is not supporting clinical evidence. Please, rewrite.
  • Abstract (line 10): change into “a labdane diterpene”
  • Introduction (line 34): the EMA assessment reported should be included (EMA/HMPC/604273/2012).
  • Introduction (line 34-40): the true impact and novelty of this review should be highlighted.
  • Morphology should be changed into “Botanical features”
  • Further paragraphs describing “cultivation and plant growth” and “Post-harvesting drug processing” can be included after “Botanical features”. The paragraphs Growing (lines 400-443) and 11. Post-harvesting processing (lines 444-483) should be moved here.
  • Chemistry should be changed into “Phytochemical composition”
  • Chemistry (line 62). Please move here the wording at pag 14 lines 407-408 “Marrubii herba (leaves and tops) are harvested just before full green color. M. vulgare has musky 407 odor which changes by drying into pungent bitter, yet pleasant and aromatic taste [2].”
  • The bitterness value of Marrubium vulgare herba should be included (see EMA – assessment report), also discussing if it could represent a limit for the use.
  • Table 1: please, specify “du”
  • Table 1: the number of decimal places should be the same for all the values
  • Medicinal properties: the type of biological study should be explained. This reviewer thinks that different paragraphs including “In vitro biological activities” and “In vivo biological activities” could be more appropriate. Also, a further paragraph concerning clinical evidence should be included.
  • 2 Anticancer properties should be changed into “Antiproliferative activity”
  • 4. Antidiabetic activity: improve this paragraph taking into account the following study “Rodríguez Villanueva J, Martín Esteban J, Rodríguez Villanueva L. A Reassessment of the Marrubium Vulgare L. Herb's Potential Role in Diabetes Mellitus Type 2: First Results Guide the Investigation toward New Horizons. Medicines (Basel). 2017;4(3):57.”
  • Lines 401-406: this point can be deleted
  • Line 701 “Sample Availability: Samples of the compounds ...... are available from the authors.” : check and correct

Author Response

Dear reviewer,

Thank you for the highly constructive comments. We made the effort to modify the paper.

We change title to: Marrubium vulgare: structure diversity with multi-functionality in the phytochemistry.

Having in mind that almost all reviewers advised rearrangement of subtitles (Please see the attachment).

Sincerely

Autors

Reviewer 4 Report

I read carefully the proposed manuscript for publication titled 'Marrubium vulgare - chemistry and bioactivity ". Particular emphasis was given on the review references 2, 3, and 4, in which the respective authors dealt also with the Marrubium vulgare plant.
Indeed, herein the authors presented important information regarding the biological activity of the plant. Nevertheless, in contrast to the above, while the title of the manuscript clearly refers to the chemical profile of the plant, although the manuscript contains a detailed Table (Table 1) presenting phytochemicals (mainly nonvolatile) of Marrubium vulgare extracted from about thirty references, only a few of them(nine) are adequately discussed in the text. In conclusion, the part of the manuscript that refers to phytochemistry of Marrubium vulgare plant is not equally discussed. Therefore, my final decision is major revision.

Author Response

Dear reviewer,

Thank you for the highly constructive comments. We made the effort to modify the paper. 

Having in mind that almost all reviewers advised rearrangement of subtitles, the new subtitles please see at the attachment.

We change title to: Marrubium vulgare: structure diversity with multi-functionality in the phytochemistry. After significant changes, the subtitle 3. Phytochemical composition is significantly improved.

Sincerely,

Autors

Reviewer 5 Report

Dear Authors, having read your manuscript I have the following comments:

  •  throughout the text, please, use from time to time the common names of Marrubium vulgare, like white horehound or common horehound
  • in the description of plant's chemistry, please give some examples of the major components of the extract. Now the authors speak generally about the groups of secondary metabolites (like alkaloids, saponins, etc.) and they present the full list of compounds in the table 2. It would be nice to read about some of the major constituents in the text
  • line 116: the acetylcholinesterase inhibitory activity determination should not be here in this section on the antioxidnt activity determination? please remove these data from here
  • in the descriptions of the biological activity horehound the Authors speak about the activity of the total extracts, like it is in the case of antimicrobial activity. Please, try to enrich your work with the data on the biological potential of single compounds  that come from Marrubium
  • please, move the chapter 5 above before chapter 4
  • the Authors should prepare a graph illustrating the pharmacological potential of the plant
  • please, add a chapter 'perspectives' at the end of the work
  • i would advise the Authors to include the data on the sources that were used - how many records were searched from Scopus that desrcibe marrubium? is the study on this plant species popular, are there many publications on this topic? please comment on that

other minor comments:

  • line 26: please change have to has
  • - table 1: what is 'du'?

Author Response

Dear reviewer,

we thank you for highly constructive suggestions which have significantly contributed to quality of this paper.

After significant changes, the subtitle 3. Phytochemical composition is significantly improved. Table 2 is divided into 3 tables according to groups of secondary metabolites. Having in mind that almost all reviewers advised rearrangement of subtitles, the new subtitles are included.

However, we maintained Marrubium vulgare (M. vulgare) throughout the text for uniformity purpose.

This review article was written during the research of M. vulgare collected in a few consecutive years. The aim was to establish the effect of climatic conditions on the chemical composition and its biological activity. However, biological potential of single compounds that come from Marrubium will be main topic in some of our future papers.

Sincerely

Autors

Round 2

Reviewer 1 Report

The authors followed the reviewers comments and suggestions and improved their manuscript, so it is suitable for publication.

Author Response

Dear reviewer,

The authors would like to thank you for a quick and professional review. It is obvious that you are expert in this field. All your remarks are accepted and paper is changed according to comments.

Best regards

Authors

Reviewer 2 Report

RE: Revision review -molecules-836764

Molecules (ISSN 1420-3049);

In the revision manuscript, the authors now show convincingly along with quality improvement of description regarding Marrubium vulgare replacing with changing subtitles and correct some contexts along with crosscheck reference we concerned was defined clearly. Hence, I believe it is ready to publish under the title which may be interesting to the community of science those who are trying to do therapeutic application in various diseases. Overall this is a much better manuscript and shows the impact of M. Vulgare as a multifunctional natural compound with structural diversity in their photochemical structure.

Author Response

(The authors gave the same response as above.)

Reviewer 3 Report

The Authors improved partially the manuscript, however some key points require to further assessed to make it suitable for publication. 

  • The new title not clearly depict the content of the review
    Please, consider the following alternative 
    “Marrubium vulgare L.: a phytochemical and pharmacological overview”
  • Abstract - "good antimicrobial properties": a good activity requires a comparison with standard effective compounds. Please, change. 
  • Keywords should be better focused
  • "Botanical features": please change into "Botanical and agronomical features" which better depict the paragraph content
  • As stated in my previous comments, a further paragraph concerning clinical evidence should be included.
  • The “Conclusions” paragraph should be improved: the future impact of this plant in the field of natural pesticides or in human medicines, also taking into account clinical efficacy, should be discussed.
  • Some typing and formatting errors should be checked and corrected.

Author Response

(The authors gave the same response as above.)

Reviewer 4 Report

Accept in present form

Author Response

(The authors gave the same response as above.)

Reviewer 5 Report

Dear Authors,

thank you for the revision of your manuscript.

I have no more concerns

Author Response

(The authors gave the same response as above.)
